# Personality Traits Predict Life Satisfaction in Coronary Heart Disease (CHD) Patients

**DOI:** 10.3390/jcm11216312

**Published:** 2022-10-26

**Authors:** Weixi Kang

**Affiliations:** UK DRI Care Research and Technology Centre, Department of Brain Sciences, Imperial College London, London W12 0BZ, UK; weixi20kang@gmail.com

**Keywords:** personality, big five, life satisfaction, CHD

## Abstract

Objectives: The objective of the current study is to establish the association between Big Five personality traits and life satisfaction in coronary heart disease (CHD) patients. Methods: The current study analyzed data from 566 patients with CHD with a mean age of 63.00 ± 15.23 years old (61.13% males) and 3018 healthy controls (63.95% females) with a mean age of 63.85 (S.D. = 9.59) years old from the UKHLS. A train-and-test approach accompanied by one-sample t-tests was used to analyze the differences in personality traits and life satisfaction between CHD patients and healthy controls while controlling for potential confounders. Two multiple regression models were applied to analyze the associations between personality traits and life satisfaction in CHD patients and healthy controls, respectively. Results: The current study found that CHD patients have lower conscientiousness scores than healthy controls. Moreover, neuroticism was negatively related to life satisfaction, and agreeableness, conscientiousness, and extraversion were positively related to life satisfaction in healthy controls. However, only neuroticism and agreeableness were related to life satisfaction in CHD patients. Conclusion: Health professionals and clinicians should utilize findings from the current study to make customized interventions based on CHD patients’ personality traits to gain better well-being outcomes such as life satisfaction.

## 1. Introduction

Although there have been improvements in cardiovascular disease outcomes in recent years, cardiovascular diseases are still the leading cause of morbidity and death across the globe. In Europe, there is an estimated number of 3.9 million deaths and a cost of EUR 210 billion a year [1]. Among these cardiovascular diseases, coronary heart disease (CHD) is a chronic condition where the heart’s blood supply is interrupted or blocked by the accumulating fatty substances in the coronary arteries, which contributes to a substantial economic burden on the healthcare system [2].

Measuring life satisfaction is regarded as an advantageous way of assessing overall well-being, given that life satisfaction represents “a cognitive and global evaluation of one’s life as a whole”, which is quite stable over time [3,4,5]. Moreover, life satisfaction is also determined by many factors, including genetic predisposition, personality traits [6], major life events, cultural background [7], and sociodemographic variables, such as age, sex, educational level, marital status, and income [6,8]. In addition, psychological well-being encompassing constructs such as life satisfaction has emerged as an important public health target [9,10,11,12]. Life satisfaction can also be improved by interventions [9] and rehabilitation [13] in CHD patients. Life satisfaction is also associated with a lower risk of cardiovascular disease [14,15,16,17] and pre-disease stages of cardiovascular disease [18,19,20]. Dag et al. (2022) found that life satisfaction was associated with less coronary artery calcification [21]. Boehm et al. (2011) found that positive psychological well-being is associated with a consistently reduced risk of CHD after controlling for cardiovascular risk factors and ill-being in males and females [14].

Two theories of life satisfaction have been discussed intensively in the literature, which are the bottom-up and top-down theories [22,23,24]. Specifically, the bottom-up theory considers overall life satisfaction as a function of satisfaction in various aspects of life, such as satisfaction with leisure, job, and health [7]. Individuals’ responses to questions about their life are not straightforward averages of domains of life satisfaction but may be determined by how people assess domains differently [7]. For instance, while some people consider a job the most important aspect of their lives, others may evaluate health. Satisfaction with the aspect of life that is in line with one’s values has been shown to be the most critical for one’s overall life satisfaction [25]. Moreover, complaints in one domain may lead people to reevaluate that domain’s relevance toward total life satisfaction [26]. For example, one may put a greater emphasis on family contentment for someone who suffers from health issues such as CHD. On the other hand, the top-down theory considers overall life satisfaction as a result of personality traits and other stable characteristics [22,27]. According to this view, life satisfaction is determined by personality disposition, which refers to the basic and stable individual differences in terms of how a person thinks, feels, and behaves. This theory has been extensively examined using models such as the Big Five, which include neuroticism (i.e., the tendency to experience negative emotions rather than being emotionally stable), agreeableness (i.e., the tendency to be polite and cooperative rather than being rude and antagonistic,) openness (i.e., the tendency to be open to new experiences rather than being conservative), conscientiousness (i.e., the tendency to be task-focused and orderly rather than being disorganized), and extraversion (i.e., the tendency to be assertive and sociable). Studies have examined the association between Big Five personality traits and life satisfaction and have yielded controversial results, with neuroticism [7,28,29] negatively associated with overall life satisfaction, and openness [7,28,29], agreeableness [28,29], conscientiousness [7,28,29], and extraversion [28,29] positively associated with life satisfaction. More recently, Malvaso and Kang (2022) [7] and Lachmann et al. (2017) [28] argued that rather than a unitary bottom-up or top-down theory alone, life satisfaction should be considered as the interplays between sociodemographic factors, areas of life satisfaction, and other stable characteristics such as personality traits.

Thus, one may expect that CHD may affect the link between personality and life satisfaction. Moreover, this research question is important given that life satisfaction is an important indicator of well-being, and health professionals are finding ways to improve well-being and outcomes in patients. The aim of the current study is to look at how Big Five personality traits are related to life satisfaction in CHD patients.

## 2. Materials and Methods

### 2.1. Data

The study used data from Understanding Society: the UK Household Longitudinal Study (UKHLS), which has been collecting annual information from the original sample of UK households since 1991 [30]. This dataset is publicly available at https://www.understandingsociety.ac.uk/documentation/mainstage (accessed 10 September 2022). All data collections have been approved by the University of Essex Ethical Committees. Participants received informed consent before participating in these studies. All participants first answered the question regarding if they had been clinically diagnosed with CHD in Wave 1 (collected between 2009 and 2010). Then, participants were asked again if they had been newly diagnosed with CHD at each wave until Wave 3. In addition, participants completed questionnaires regarding personality, demographic, and psychological distress questions at Wave 3 (collected between 2011 and 2012). Age and sex-matched controls were selected from the people who had not been clinically diagnosed with CHD. Thus, there were 566 patients with CHD with a mean age of 63.00 ± 15.23 years old (61.13% males) and 3066 healthy controls (63.95% females) with a mean age of 63.58 ± 9.59 years old.

### 2.2. Measures

#### 2.2.1. CHD

The validity of self-reported cardiovascular disease has been approved (e.g., [31]). Retrospectively, CHD was measured by the question, “Has a doctor or other health professional ever told you that you have any of these conditions? Coronary heart disease.” at Wave 1. In follow-up waves, participants were asked if they were newly diagnosed with CHD at each wave.

#### 2.2.2. Big Five Personality Traits

Personality was measured using the 15-item version of the Big Five Inventory [32] with a Likert scale ranging from 1 (“disagree strongly”) to 5 (“agree strongly”). Scores were reverse-coded when appropriate. The exact set of questions used to ask participants can be found at https://www.understandingsociety.ac.uk/documentation/mainstage/dataset-documentation/term/personality-traits?search_api_views_fulltext= (accessed 10 September 2022). Exemplary items include “I see myself as someone who worries a lot” (neuroticism), “I see myself as someone who is sometimes rude to others.” (agreeableness), “I see myself as someone who is original, comes up with new ideas” (openness), “I see myself as someone who does a thorough job” (conscientiousness), and “I see myself as someone who is talkative” (extraversion).

#### 2.2.3. Life Satisfaction

Participants answered the question “How dissatisfied or satisfied are you with… your life overall?” using a 7-point scale ranging from 1 (not satisfied at all) to 7 (completely satisfied). The results of single-item measures and multi-item measures, such as the Satisfaction with Life Scale (SWLS), have been shown to be very similar [33].

#### 2.2.4. Control Variables

Demographic variables included age, sex, monthly income, the highest educational qualification, marital status, and psychological distress as measured by the GHQ-12 (Goldberg & Williams, 1988). Specifically, age, monthly income, and psychological distress were coded as what they were (continuous), sex was coded as male (1) vs. female (2), the highest educational qualification was coded as below college (1) vs. college (2), and marital status was coded as (1) single vs. (2) married.

### 2.3. Analyses

First, the current study looked at the differences in personality traits and life satisfaction between CHD patients and age and sex-matched healthy controls. In order to do so, four generalized linear models were first trained based on demographics and psychological distress data on healthy controls; then, demographics and psychological distress from CHD patients were taken into these models to predict expected scores in CHD patients given their demographics and psychological distress. Finally, one-sample t-tests were applied to compare the differences in Big Five and life satisfaction scores between the predicted and expected scores. This approach is more advantageous than the pairwise sample t-test as it controls for other possible confounders other than age and sex in the current case. Finally, two multiple regressions were then used to predict life satisfaction by taking demographics, personality traits, and psychological distress as the predictors to predict life satisfaction in CHD patients and healthy controls, respectively. All above-mentioned analyses were conducted in MATLAB 2018a.

## 3. Results

The descriptive statistics of CHD patients can be found in Table 1. The current study found that CHD patients have significantly lower conscientiousness (t(565) = −3.98, *p* < 0.01, 95% C.I. (−0.27, −0.10), Cohen’s d = −0.17) scores compared to age and sex-matched healthy controls after controlling for other demographics and psychological distress. The regression model explained 25% (R^2 = 0.25) of the total variance in life satisfaction in healthy controls, with neuroticism (b = −0.06, *p* < 0.001, 95% C.I. (−0.08, −0.03)) negatively related to life satisfaction and agreeableness (b = 0.05, *p* < 0.001, 95% C.I. (0.02, 0.08)), conscientiousness (b = 0.06, *p* < 0.001, 95% C.I. (0.04, 0.09)), and extraversion (b = 0.04, *p* < 0.001, 95% C.I. (0.02, 0.06)) positively related to life satisfaction. However, in CHD patients, the regression model explained 30.1% (R^2 = 0.301) of the total variances. Neuroticism (b = −0.10, *p* < 0.05, 95% C.I. (−0.19, −0.01)) was negatively related to life satisfaction, whereas agreeableness was positively related to life satisfaction (b = 0.13, *p* < 0.05, 95% C.I. (0.03, 0.23)). The results of the full regression model can be found in Table 2.

## 4. Discussion

The aim of the current study was to look at the associations between personality traits and life satisfaction in CHD patients. The current study analyzed data from 566 patients with CHD with a mean age of 63.00 ± 15.23 years old (61.13% males) and 3018 healthy controls (63.95% females) with a mean age of 63.91 ± 9.62 years old from the UKHLS. The current study found that CHD patients have lower conscientiousness scores compared to healthy controls. Moreover, neuroticism was negatively related to life satisfaction, and agreeableness, conscientiousness, and extraversion were positively related to life satisfaction in healthy controls, which seems to be mostly consistent with the previous literature on healthy participants [7,28,29]. However, only neuroticism and agreeableness were related to life satisfaction in CHD patients.

The associations between CHD and Big Five personality traits are not yet well established in the literature. However, previous studies have found that Type A personality, which refers to the tendency to be competitive, aggressive, hostile, and time-pressured [34,35,36,37,38], and Type D, which refers to the tendency to have high negative affectivity combined with high levels of social inhibition, are the major risk factors for CHD [39,40]. The fact that CHD patients had lower conscientiousness scores can be explained by the fact that CHD may not allow physically disabled people to stick to their tasks and goals. Moreover, conscientiousness is related to health-promoting behaviors such as physical activity [41,42], stronger grip strength, better lung function and faster walking speed [43], fewer harmful behaviors, such as smoking and alcohol use [44,45], fewer risks of chronic conditions [46] such as obesity [47], and few depressive symptoms [48]. Biological factors may also explain this link. Specifically, conscientiousness is associated with healthier metabolic, cardiovascular, and inflammatory markers [43,49] and higher cardiorespiratory fitness [50].

Neuroticism refers to the tendency to experience negative emotions rather than being emotionally stable. The finding that neuroticism is negatively related to life satisfaction in CHD patients is largely consistent with the literature [7,28,29]. People who score high in neuroticism may tend to perceive the world in a negative way. In addition, neuroticism is also related to self-rated health, behavioral markers, such as sleep quality and walking speed [51,52], biological dysfunction [53], and poor health outcomes, such as chronic respiratory diseases, major depression, and dementia [48,54,55]. These clinical conditions may, in turn, result in poor life satisfaction.

The finding that agreeableness is positively associated with life satisfaction is consistent with previous research [28,29] and the notion that agreeable people may have a higher chance to uphold social conversions that are conducive to behaviors that benefit health than disagreeable individuals [56]. Thus, agreeable CHD patients may be more likely to follow instructions and advice than disagreeable CHD patients after diagnosis of CHD, which may then benefit psychological well-being as measured by life satisfaction. Moreover, agreeable CHD patients may be able to develop and maintain high levels of social support after CHD diagnosis. Indeed, studies have found that agreeableness predicts the quality of friendships [57] and can facilitate higher-quality relationships compared to CHD counterparts, which may, in turn, have positive impacts on psychological distress and health-related behaviors [58]. Agreeableness is also related to active coping [59]. Moreover, agreeable individuals may have better coping styles than disagreeable people [60]. The author hopes that the current finding can inspire new research aiming to explain these effects by identifying potential mediators.

Despite the strength of the strengths of the current study, including a relatively large sample size and well-controlled sociodemographics, there are still some limitations. First, the current study was based on cross-sectional data, which makes it hard to identify the causal relationship. Future research should use longitudinal designs to try to identify the causal relationship. Second, the current study was based on self-reported measures, which cannot avoid self-reporting bias. Thus, future studies should use more objective measurements to check if the results from the current study still hold. Finally, the current study focused on participants from the United Kingdom, which may make it hard to generalize the current finding to other countries. Future studies should investigate populations from other countries.

Taken together, the current study investigated how Big Five personality traits relate to CHD patients. The current study found that CHD patients have lower conscientiousness scores than healthy controls. Moreover, neuroticism was negatively related to life satisfaction, and agreeableness, conscientiousness, and extraversion were positively related to life satisfaction in healthy controls. However, only neuroticism and agreeableness were related to life satisfaction in CHD patients. There are also some implications of the current study. Health professionals and clinicians should utilize findings from the current study to make customized interventions based on CHD patients’ personality traits to gain better well-being outcomes, such as life satisfaction.

## Figures and Tables

**Table 1 jcm-11-06312-t001:** Descriptive statistics for healthy control and CHD patients.

	Healthy Controls (N = 566)		CHD Patients (N = 3018)	
	Mean	S.D.	Mean	S.D.
Age	63.94	9.70	63.00	15.23
Monthly income	1560.09	1687.25	1380.76	1112.09
Life satisfaction	5.26	1.50	2.51	1.18
GHQ-12	10.52	5.00	11.86	5.83
Neuroticism	3.25	1.47	3.49	1.50
Agreeableness	5.65	1.05	5.57	1.14
Openness	4.47	1.36	4.33	1.44
Conscientiousness	5.52	1.13	5.32	1.18
Extraversion	4.48	1.37	4.49	1.34
	**N**	**%**	**N**	**%**
**Sex**				
Male	1934	63.08	346	61.13
Female	1132	36.92	220	38.87
**Highest educational qualification**				
Below college	2274	74.17	441	77.92
College	791	25.83	125	22.08
**Legal marital status**				
Single	1053	34.34	224	39.58
Married	2013	65.66	342	60.42

**Table 2 jcm-11-06312-t002:** The estimates (*b*) of multiple regression models for healthy controls and people who have been diagnosed with CHD by taking demographics, psychological distress, and personality traits as the predictors and life satisfaction as the predicted variable. All numbers are rounded up to two digits.

	Healthy Controls	CHD Patients
Age (continuous)	0.02 ***	0.00
Sex (1 = male, 2 = female)	0.19 ***	0.11
Monthly income (continuous)	0.00 ***	0.00 **
Highest educational qualification (1 = below college, 2 = above college)	0.10 **	−0.07
Legal marital status (1 = single, 2 = married)	0.26 ***	−0.02
Psychological distress (continuous)	−0.12 ***	−0.12 ***
Neuroticism	−0.06 ***	−0.10 *
Agreeableness	0.05 ***	0.13 *
Openness	0.02	0.03
Conscientiousness	0.06 ***	0.07
Extraversion	0.04 ***	0.03
R^2	0.25	0.30

* *p* < 0.05 ** *p* < 0.01 *** *p* < 0.001.

## Data Availability

Publicly available datasets were analyzed in this study. This data can be found here: https://www.understandingsociety.ac.uk.

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
