# Peer review of "Personality Traits Predict Life Satisfaction in Coronary Heart Disease (CHD) Patients"

_jcm, 2022, doi:10.3390/jcm11216312_

Round 1

Reviewer 1 Report

Kang performed an interesting study, my big compliment to the author! 

There are however some issues in statistical methodology

Table 2: please add the units in the effect was calculated for. For example, Income- Effect per 1 Fund? Highest educational qualification, what is the unit and what is the reference? Highest compared to lowest. Legal marital status, what is the unit?  Married compared to single?  Regression models always need reference groups. This should be explained

Further points:

Line 105-106 I cannot access to the internet page https://www.understandingsociety.ac.uk/documentation/mainstage/dataset-documentation/term/personality-traits?search_api_views_fulltext=

I would prefer to see a little bit more description of Big Five Personality traits (maybe 2-3 sentences where traits are listed) Authors first call them in results but they first should be called in methods.

Table 1: Please add the total number of people (Health controls ,N=XX; CHD, N=XX)

Author Response

Thanks so much for reviewing my manuscript. I have now addressed these minor issues regarding tables/exemplary items of Big Five inventory with tracked changes. In addition, I can confirm that the link should point to the correct destination. 

Reviewer 2 Report

This manuscript describes an investigation into the factors associated with lifestyle satisfaction in CHD patients. Based on t-test analyses, it was observed that in comparison to non-CHD controls, the CHD patients were lower in conscientiousness.  Based on regression analyses, there was a negative relationship between Neuroticism and life satisfaction and a positive relationship between Agreeableness and life satisfaction in the CHD patients. A somewhat different pattern was observed in non-CHD patients. The results are described as being fairly consistent with logical expectations and the existing literature.

This manuscript will likely be of interest to readers of the journal. It flows logically and is for the most part easy to read and understand. I have only minor suggestions prior to publication,

1.       Please double-check the entire document for grammatical errors and errors and syntax. An example of a grammatical problem appears on line 24 “In Europe, the is an estimated number…” instead of “In Europe, THERE is an estimated number…” An example of syntax problems would be the frequent failure to close parentheses in the paragraph that starts on line 43.

2.       Please delete the first sentence of the last paragraph in the Introduction because since the data on CHD and personality was discussed in a paragraph that did not immediately precede this one, the sentence is a little confusing.  Delete it and start the next sentence with “This research is important given that life satisfaction…”

3.       Please reword the sentence (or section) of the Discussion section which contains the statement that “CHD patients had less Conscientiousness scores can be explained by the fact that CHD may allow physically disabled people to stick to their tasks and goals.” This is contrary to logic.  Why would a disease allow people to better stick to their tasks?

Otherwise, I have no issues with this manuscript and feel that it should be published without further review once these changes are made.

Author Response

Thanks so much for reviewing my manuscript. I have now addressed these minor issues regarding the wording/grammar errors with tracked changes.